# Genotypic and Haplotypic Association of Catechol-*O*-Methyltransferase rs4680 and rs4818 Gene Polymorphisms with Particular Clinical Symptoms in Schizophrenia

**DOI:** 10.3390/genes14071358

**Published:** 2023-06-27

**Authors:** Marina Sagud, Lucija Tudor, Gordana Nedic Erjavec, Matea Nikolac Perkovic, Suzana Uzun, Ninoslav Mimica, Zoran Madzarac, Maja Zivkovic, Oliver Kozumplik, Marcela Konjevod, Dubravka Svob Strac, Nela Pivac

**Affiliations:** 1Department for Psychiatry and Psychological Medicine, University Hospital Centre Zagreb, 10000 Zagreb, Croatia; marinasagud@mail.com (M.S.); zoranmadzarac@gmail.com (Z.M.); maja.zivkovic@kbc-zagreb.hr (M.Z.); 2School of Medicine, University of Zagreb, 10000 Zagreb, Croatia; suzana.uzun@gmail.com (S.U.); nino.mimica@gmail.com (N.M.); 3Laboratory for Molecular Neuropsychiatry, Division of Molecular Medicine, Ruder Boskovic Institute, 10000 Zagreb, Croatia; lucija.tudor@gmail.com (L.T.); gnedic@irb.hr (G.N.E.); mnikolac@irb.hr (M.N.P.); marcela.konjevod@irb.hr (M.K.); dsvob@irb.hr (D.S.S.); 4Department for Biological Psychiatry and Psychogeriatrics, University Psychiatric Hospital Vrapce, 10090 Zagreb, Croatia; okozumplik@hotmail.com; 5University of Applied Sciences Hrvatsko Zagorje Krapina, 49000 Krapina, Croatia

**Keywords:** COMT, clinical symptoms, cognition, haplotype, polymorphisms, schizophrenia, sex-differences

## Abstract

Catechol-*O*-methyl transferase (*COMT*) gene variants are involved in different neuropsychiatric disorders and cognitive impairments, associated with altered dopamine function. This study investigated the genotypic and haplotypic association of *COMT* rs4680 and rs4618 polymorphisms with the severity of cognitive and other clinical symptoms in 544 male and 385 female subjects with schizophrenia. *COMT* rs4818 G carriers were more frequent in male patients with mild abstract thinking difficulties, compared to CC homozygotes or C allele carriers. Male carriers of *COMT* rs4680 A allele had worse abstract thinking (N5) scores than GG carriers, whereas AA homozygotes were more frequent in male subjects with lower scores on the intensity of the somatic concern (G1) item, compared to G carriers. Male carriers of *COMT* rs4818–rs4680 GA haplotype had the highest scores on the G1 item (somatic concern), whereas GG haplotype carriers had the lowest scores on G2 (anxiety) and G6 (depression) items. *COMT* GG haplotype was less frequent in female patients with severe disturbance of volition (G13 item) compared to the group with mild symptoms, while CG haplotype was more frequent in female patients with severe then mild symptoms. These findings suggest the sex-specific genotypic and haplotypic association of *COMT* variants with a severity of cognitive and other clinical symptoms of schizophrenia.

## 1. Introduction

Schizophrenia, one of the most severe mental disorders, is associated with the imbalance between resilience and vulnerability in relation with life stressors [1,2,3]. Schizophrenia is characterized by positive, negative and cognitive symptoms, according to the DSM-5 criteria [4]. Cognitive symptoms include disturbances in executive function, working and verbal memory, language, learning, attention, processing speed, vigilance, and problem solving [4,5,6,7], leading to decreased social functioning and adjustment and employment problems [4,5,7,8]. Cognition is affected by different neurotransmitter systems [9], with the important role of various proteins and genes [10].

Dopamine is associated with neurobiological underpinning of schizophrenia [11], and with cognitive processing [12,13]. Catechol-*O*-methyl transferase or COMT is an enzyme that degrades dopamine, noradrenaline, and adrenaline, and acts as an important modulator of the brain function, especially in the prefrontal cortex [14,15], the brain region engaged in most of the cognitive processes. COMT is an enzyme involved in the inactivation of compounds having a catechol structure, including catecholamine neurotransmitters such as dopamine, by introducing a methyl group, donated by S-adenosyl methionine, to the catecholamine. The active site of COMT consists of the S-adenosyl-l-methionine (SAM) binding domain and the catalytic site. The catalytic site contains a metal ion (Mg^2+^) and amino acids important for substrate binding and catalysis of the methylation reaction. The polymorphic residue according to rs4680 is buried in a hydrophobic residue, around 16A° away from the SAM-binding site [14], and in a complementary hydrophobic methyl binding pocket according to rs4818 [15].

One of the most frequently investigated genes, associated with cognitive phenotypes, is the *COMT* gene. Therefore, *COMT* gene and its polymorphisms have been studied in different neuropsychiatric disorders and cognitive impairments, associated with altered dopamine function [16]. The most often studied is a functional polymorphism *COMT* Val158Met (rs4680) [17].

Specifically, *COMT* rs4680 polymorphism, or a G/A substitution, leads to valine’s (Val) replacement with methionine (Met) at codon 158 of the membrane-bound COMT (MB-COMT) and at codon 108 of the soluble short form (S-COMT) [18], and results in a significant (three- to four-fold) fall in the COMT activity in the A (Met) carriers. Another *COMT* polymorphism, *COMT* rs4818 polymorphism is located on exon 4, and consists of a C/G substitution (Leu/Leu) at codon 86 of the S-COMT and at codon 136 of the MB-COMT [19]. Since GG carriers have higher COMT activity than the CC carriers of this polymorphism, presence of the G variant is related to greater COMT activity and reduced prefrontal dopamine activity [19]. It is assumed that COMT activity is more under influence of the *COMT* rs4818 than *COMT* rs4680 polymorphism [20]. Genotypic [21,22] and haplotypic [23] associations of the *COMT* were shown for particular features in schizophrenia. Besides GWAS, that need thousands of samples and multicenter studies [24], haplotype-based studies might detect association or modulation of specific domains of symptoms of schizophrenia, compared to negative association studies using a single SNP or individual genotypes related to *COMT* rs4680 [25] or *COMT* rs4818 [26,27]. Haplotype associations have been reported for anhedonia [28], and for the treatment response [29,30].

Inconsistent findings across studies might also be associated, among many other factors, with the sexually dimorphic influence of *COMT* upon various psychiatric symptoms [31] and sex related differences in schizophrenia [28].

In the present study, we controlled *COMT* rs4680 and rs4818 polymorphisms data for the possible effects of sex, age, and smoking, and evaluated all 929 subjects (544 male and 385 female participants) with chronic schizophrenia in unrelated Caucasian subjects, using the Positive and Negative Syndrome Scale (PANSS) [32]. We determined this association with PANSS cognition subscale [33] symptoms (i.e., with P2 = Conceptual disorganization, N5 = Abstract thinking, G10 = Disorientation, G11 = Attention problems) [34], The PANSS symptom level deconstruction established the utility of the use of the selective cognitive items and other negative or general psychopathology items listed from the PANSS [35].

Regarding genetic models, we assessed an association of symptoms in schizophrenia with *COMT* rs4680 or *COMT* 4818 polymorphisms using genotypic, allelic, dominant (AA + GA vs. GG for *COMT* rs4680 and CG + GG vs. GG for *COMT* rs4818) and recessive (AA vs. GA + GG for *COMT* rs4680 and CC vs. CG + GG for *COMT* rs4818) models, respectively. We expected that *COMT* variants would be associated with cognitive, negative or, general PANSS psychopathology in our patients. We hypothesized that the presence of the G allele of the *COMT* rs4680 or rs4818 polymorphism, both related to higher COMT activity and reduced dopaminergic function, will be associated with pronounced cognitive decline and more severe clinical symptoms in schizophrenia, compared to presence of the A or C allele of the *COMT* rs4680 or rs4818 polymorphisms, respectively.

## 2. Materials and Methods

### 2.1. Subjects

The study included 929 subjects (544 male and 385 female participants) with schizophrenia, and some of them were included in our previous studies [28,29,30]. Patients were included in the study if they had schizophrenia for at least 5 years, and if they were 18–65 years old. Before participation, patients were informed in detail about the procedures, and after they had signed the written informed consent, the study was carried out in accordance with The Code of Ethics of the World Medical Association (Declaration of Helsinki from 1975). Both in- and out-patients participated. Exclusion criteria were intellectual disabilities, first-episode psychosis, substance abuse and/or dependence in the previous three months, and if they had any comorbid severe somatic or neurological disorder.

They were diagnosed using the Structured Clinical Interview for DSM-IV (SCID) [36], while the severity of clinical symptoms was evaluated using the gold standard, the Positive and Negative Syndrome Scale (PANSS) [32], and the presence of cognitive decline was determined with PANSS cognitive subscale, that includes items P2 (conceptual disorganization), N5 (abstract thinking), G10 (disorientation), and G11 (attention problems) [33,34]. Additionally, subjects were subdivided into those with mild and moderate to severe symptoms with cut off points: 4 for particular PANSS items, i.e., 16 for PANSS cognitive subscale, 28 for PANSS positive and negative subscale, and 64 for PANSS general subscale, as reported previously [37]. Average PANSS score was 103 (88; 124). All subjects were recruited from the University Hospitals from the Zagreb County (Department for Psychiatry and Psychological Medicine at the University Hospital Centre Zagreb and from Department for Biological Psychiatry and Psychogeriatrics, University Psychiatric Hospital Vrapce, Zagreb, Croatia). The study was conducted with the approval of the Ethics Committees of the University Hospital Center Zagreb and University Psychiatric Hospital Vrapce, Zagreb, Croatia.

### 2.2. Genotyping

All subjects were sampled during regular check-ups in the morning, and blood was processed on the same day. Genomic DNA was isolated from the peripheral blood using the salting-out method [38]. The *COMT* rs4818 (assay ID: C__2538750_10) and *COMT* rs4680 (assay ID: C__25746809_50) polymorphisms genotypes were determined using the primers and probes from Applied Biosystems as TaqMan^®^ Drug Metabolism Genotyping Assays (Applied Biosystems, Foster City, CA, USA) on ABI Prism 7300 Real time PCR System apparatus (Applied Biosystems, Foster City, CA, USA), according to the procedures described by Applied Biosystems. Thermal cycler conditions were 10 min at 95 °C followed by 50 cycles of denaturation at 92 °C for 15 s and elongation at 60 °C for 90 s. The 10 μL reaction volume contained around 30 ng of DNA. Besides the codominant model, which includes all three genotypes, dominant, and recessive models for both *COMT* polymorphisms, as well as the allelic model, were used.

### 2.3. Statistical Analyses

The data were analyzed using Prism version 7.00 (GraphPad Software, Inc., San Diego, CA, USA). Due to the deviation from a normal distribution (determined with Kolmogorov–Smirnov test), nonparametric tests were used for all analyses. Multiple linear regression was performed to analyze the effect of sex, age, and smoking on severity of cognitive decline (PANSS cognitive scores) and severity of other individual items of the PANSS in patients with schizophrenia. The Mann–Whitney U test was used for independent pair comparisons and Kruskal–Wallis ANOVA, with post hoc Dunn test was used for multiple comparisons. Box-plot diagrams were used for visual representation of significant results, where the central box represented the interquartile range, the middle line represented the median, the vertical line extended from the minimum to the maximum value, while separate dots represented the outliers. Extreme values (more than 3 box-lengths outside of the box) were excluded from the analyses. Genotype, allele, and haplotype distribution among subjects with mild and severe symptoms of schizophrenia was calculated using the χ^2^-test, while the standardized residuals (R) were calculated to determine which parameter was the most significant contributor to the differences among groups [39].

The Hardy–Weinberg equilibrium for *COMT* rs4818 and rs4680 polymorphisms was calculated using χ^2^-test [40]. LD pairwise values for two *COMT* SNPs and haplotype frequencies were determined using the Haploview software v. 4.2 [41] and represented with D’ value. Loci are considered to be in linkage if D’ is >0.80. PLINK v. 1.07. software was used to assign best-estimate haplotype pairs to each individual using the expectation–maximization algorithm [42].

G*Power 3.1 Software (Germany) [43] was used to determine a priori sample size. For all two tailed analyses, the *p*-value (0.05/2 = 0.025) was corrected because two SNPs were compared and the results were considered significant if *p* < 0.025. Required sample size (with α = 0.025; power (1 − β) = 0.800; and a small effect size = 0.15) for Mann–Whitney test was N = 416, whereas for Kruskal–Wallis ANOVA (for 3 groups, the required sample size was N = 432; and for 4 groups, the required sample size was N = 492). As the study included N = 929 participants, it had adequate sample size and statistical power.

## 3. Results

### Demographic Data

The study included a total of 929 subjects (544 male and 385 female participants) with schizophrenia. Male subjects were significantly younger (*p* < 0.001), and had more severe total (*p* < 0.001), positive (*p* < 0.001), negative (*p* < 0.001), and general (*p* < 0.001) symptoms of schizophrenia, as well as higher PANSS cognitive scores (*p* < 0.001), than female subjects (Table 1). Moreover, male subjects were more frequently smokers (69.1%, R = 2.1; *p* < 0.001) than female subjects (51.7%).

Multiple linear regression was performed to analyze the effect of sex, age, and smoking on severity of cognitive decline (PANSS cognitive scores) in patients with schizophrenia. Younger age (β = −0.211; *p* < 0.001) and male sex (β = −0.360; *p* < 0.001) were significant predictors of higher PANSS scores, while smoking (β = −0.006; *p* = 0.858) was not associated with PANSS cognitive scores (R^2^ = 0.228; F = 63.398; *p* < 0.001). Since male subjects were significantly younger than female, and there were significant differences in PANSS scores between two sexes, all analyses were performed in male and female subjects separately.

Genotype and allele frequencies of the rs4818 and rs4680 polymorphisms located in the *COMT* gene have been determined in total 929 subjects. Minor allele frequency (MAF) and corresponding Hardy–Weinberg equilibrium significance value for each polymorphism, as well as linkage disequilibrium between *COMT* rs4818 and rs4680 polymorphisms have been determined using Haploview software v. 4.2. MAFs for rs4818 and rs4680 were 0.386 (HW *p* value = 0.859) and 0.498 (HW *p* value = 0.658), respectively. Haplotype analysis showed strong LD between *COMT* rs4818 and rs4680 polymorphisms (D’ × 100 = 88; LOD > 2) (Figure 1), hence haplotypes for *COMT* rs4818 and rs4680 block were determined for each subject using expectation–maximization algorithm. The most common haplotype was CA, which was present in 48.0% of subjects, followed by GG (36.3%) and CG (13.4%) haplotypes. The rarest haplotype was GA (2.2%). 

There were no significant differences in the distribution of the *COMT* rs4818 and rs4680 genotypes, alleles, or haplotypes between male and female patients, as well as between smokers and non-smokers (Table 2).

The severity of the PANSS positive, negative, and general psychopathology symptoms of schizophrenia, and cognitive decline evaluated with PANSS cognitive subscale, that includes items P2 (conceptual disorganization), N5 (abstract thinking), G10 (disorientation), and G11 (attention problems), as well as scores on particular items of PANSS scale, were evaluated in regard to *COMT* rs4818 and rs4680 polymorphisms and *COMT* rs4818–rs4680 haplotype in male and female subjects separately.

Polymorphism *COMT* rs4818 was not associated with any of the symptoms of schizophrenia in male or in female subjects when the total scores were evaluated (Table 3 and Table 4). However, when subjects were divided according to the severity of symptoms into those with mild symptoms and those with moderate and severe symptoms (Table 5), G carriers were more often present (25.6%) in male subjects with mild difficulties in abstract thinking (item N5) compared to male CC carriers (17.2%) (R = 1.2; χ^2^ = 5.210; *p* = 0.022), or male C allele carriers (20.2%) (R = 1.6; χ^2^ = 5.042; *p* = 0.025). Additionally, nominal association of *COMT* rs4818 polymorphism and scores on G1 item (corresponding to the intensity of somatic concern) was observed, where G allele was under-represented (19.3%) in male subjects with mild symptoms (R = −1.4; χ^2^ = 4.186; *p* = 0.041) compared to C allele carriers (24.7%); however, after correction for multiple testing, this association did not reach statistical significance.

Polymorphism *COMT* rs4680 was associated with N5 (abstract thinking) scores in male subjects, where A carriers had higher scores than GG carriers (U = 23575.5; *p* = 0.023 (Table 3, Figure 2a). As shown in Table 5, when subjects were divided according to the severity of symptoms, *COMT* rs4680 AA homozygotes were more frequent (18.1%) in subjects with lower scores on G1 (somatic concern) item (R = 1.7; χ^2^ = 5.061; *p* = 0.024), compared to G carriers (23.7%). However, this association was not confirmed in female patients (Table 6).

Both *COMT* polymorphisms showed an association with the PANSS general scores in female patients when they were subdivided in mild and severe symptoms category (Table 6); however, very low number of subjects (<2) in certain groups could possibly lead to false significant results, and since they were not confirmed when total PANSS general scores were evaluated, they were considered artefact.

The carriers of the rarest *COMT* rs4818-rs4680 haplotype (GA) had the highest scores on G1 item (somatic concern), while GG haplotype had the lowest scores on G2 (anxiety) and G6 (depression) items, compared to other haplotypes in male subjects (Table 3, Figure 2b–d and Figure 3b–d). Similarly, as shown in Table 6, in female subjects the GG haplotype was less frequent (31.8%; R = −1.4) in the group with severe disturbance of volition (G13 item) compared to the group with mild symptoms (39.8%), while CG haplotype was over-represented in group with severe symptoms (18.2%; R = 1.8), compared to those with mild symptoms (11.8%) (χ^2^ = 9.356; *p* = 0.025).

## 4. Discussion

Our results have shown a significant genotypic and haplotypic association of *COMT* rs4680 and rs4818 gene polymorphisms with particular clinical symptoms evaluated by the individual PANSS subscales scores in male and female patients with schizophrenia. Our detailed analysis detected several associations between *COMT* rs4680 and rs4818 and the severity of particular individual PANSS items. In male patients with schizophrenia, the presence of the *COMT* rs4818 G allele was more often detected in those with mild “difficulties in abstract thinking” (N5), compared to CC homozygotes or C allele carriers. Regarding the other *COMT* SNP, in male patients, the presence of the *COMT* rs4680 A allele was associated with N5 scores compared to GG carriers, and the presence of the *COMT* rs4680 AA homozygous genotype was more frequently found participants with lower scores on the intensity of somatic concern (G1) item compared to G carriers. In females with schizophrenia, there were no significant associations between individual PANSS items (clinical symptoms of schizophrenia) and *COMT* rs4680 and rs4818 genetic variants. Concerning haplotype associations, male patients with schizophrenia who were *COMT* rs4818–rs4680 GA haplotype carriers had the highest scores on G1 item (somatic concern), whereas GG haplotype carriers had the lowest scores on G2 (anxiety) and G6 (depression) items. Female patients with schizophrenia had less frequent *COMT* GG haplotype associated with severe disturbance of volition (G13 item) compared to females with mild symptoms, while female patients with CG haplotype had more frequent severe then mild symptoms. All these associations were sex-specific. 

In male patients with schizophrenia, lower N5 “Difficulties in abstract thinking” scores in *COMT* rs4680 GG carriers than in A carriers, and overrepresentation of *COMT* rs4818 G carriers compared to CC polymorphism in participants with less severe N5 scores, suggest an association between the presence higher COMT activity alleles and better preservation of abstract thinking. Actually, N5 was the only PANSS-negative item which severity was related to any of the two aforementioned *COMT* polymorphisms in males. Although abstract thinking belongs to the PANSS negative symptoms, according to a meta-analysis of the results of 45 factor analyses, it was not a part of the core negative items and was actually a symptom of disorganization/cognitive dimension [44]. The inability to engage in abstract thinking is a long-known typical symptom of schizophrenia [45]. It measures proverb interpretations and abstract similarities, by detecting the relationship between object, ideas, principles, and concepts. Its impairment is expressed as concrete form of cognition and speech, such as providing more concrete responses to proverbs [46]. Many studies have investigated abstract thinking in schizophrenia. Difficulty in abstract thinking was associated with the impaired decision making [47], distorted sensory experiences [48], and EEG findings of reduced alpha 1 spectral amplitude and lower Measurement and Treatment Research to Improve Cognition in Schizophrenia (MATRICS) Consensus Cognitive Battery 5 neuro-cognitive composite score [49]. 

Interestingly, baseline higher severity of PANSS N5 item was, along with P2 and G9 items, among three baseline PANSS items which predicted treatment resistance [50]. In another study, the PANSS item “abstract thinking” accounted for an association between negative symptoms and cognition [51], and was also found, together with conceptual disorganization, hostility, and uncooperativeness, to represent important mediating roles across different communities of schizophrenia symptoms [52]. In first-episode patients, baseline N5 and N7 were lower in remitters after one year of treatment, than in non-remitters, while other negative items were not [53]. In addition, P5 “abstractive thinking” was the only PANSS-negative symptom to predict depressive trajectory [54]. 

While these studies highlight the distinct position of N5 item among other PANSS negative symptoms [50,51,52], to best of our knowledge, only one study investigated all 30 PANSS items in relation with *COMT* polymorphism. This study found no associations between any individual PANSS item in males, including N5, with the *COMT* rs4680 polymorphism [55]. However, this study included a lower number of patients, and recruited only individuals with youth-onset schizophrenia with the mean age of 32. In our study, male patients were 38 years old. In turn, the age of onset of schizophrenia influences cognitive functioning, given that patients with youth-onset schizophrenia had severe cognitive deficits, in contrast to individuals with late-onset schizophrenia who maintained relatively preserved cognition [56]. Another study has focused on *COMT* polymorphism and abstraction in patients with schizophrenia. Our results disagree with the findings of better performance on abstraction test in individuals with schizophrenia who had greater number of *COMT* rs4680 A alleles in a dose–response fashion, although this effect was small [57]. However, the aforementioned study had small sample size for genetic study, included ethnically heterogeneous outpatient population with the average illness duration of 24 years, and abstraction was measured by a subtest from The Delis–Kaplan Executive Function System [57]. While those studies [55,57] and the present trial were cross-sectional, future studies are needed to establish the associations between *COMT* functional polymorphisms longitudinally, i.e., to explore if the relationships between *COMT* rs 4680 and rs4818 is detected in the acute phase change during successful treatment. Of note, treatment with antipsychotics decreases the scores on essentially all PANSS symptom, at least in non-treatment resistant patients [58], and it was depicted that abstract thinking in several diagnostic categories, including schizophrenia, improves over time [46]. 

The unique association of lesser severity of PANSS N5 component with the high-activity *COMT* rs4680 and rs4818 alleles in our study, indirectly suggests the involvement of lower prefrontal dopamine levels in better performance in abstract reasoning in males with schizophrenia. Namely, *COMT* genotypes at SNPs rs4680 and rs4818, but not rs737865 or rs165599, were associated with altered levels of S-COMT, but not MB-COMT, in the dorsolateral prefrontal cortex (DLPFC) of subjects who died from different causes, including suicide [59]. While it was reported that reduced activity in the left frontotemporal and the right orbitofrontal cortex could be a critical region for the deficit in abstract thinking in schizophrenia [60], low COMT activity in healthy men, resulting from the presence of the A allele of *COMT* rs4680 was associated with reductions in resting blood flow in frontal regions, compared to high COMT activity groups [61]. It may be speculated that males with lower-activity *COMT* polymorphisms in our study may have had changes in cortical neural activity and perfusion, which may have contributed to deficits in abstract thinking. Given the important role of abstract thinking in schizophrenia, it remains to be determined whether males with low-activity *COMT* alleles, who had worse N5 scores, differ in the degree of improvement of deficits in abstract thinking. 

Both G2 “anxiety” and G6 “depression” items are a part of PANSS anxiety and depression dimension, according to the factor analysis [62]. Our findings suggest that high COMT activity related to low dopamine availability was associated with the lowest anxiety and depression dimension scores in males. In another study in Croatian population, *COMT* rs4680 was not associated with affective/depressive PANSS factors [63]. Our results do not agree with the findings of the higher levels of depression in patients with *COMT*rs4680 GG genotype, compared to A carriers, in patients with first-episode schizophrenia, whereby *COMT* rs4680 polymorphism moderated an association between the severity of depression and stressful life events [64]. That study had small sample size, and in both cited studies males and females were not separately analyzed [63,64]. Importantly, in patients with schizophrenia, COMT DNA methylation was inversely correlated with depressed subdomain of the PANSS, i.e., the higher the depressive symptoms, the lower the DNA methylation [65]. Future studies exploring the associations of COMT polymorphisms and depressive symptoms in schizophrenia need to consider childhood and life-time stressors, which may strongly influence this relationship. 

The disturbance of volition (PANSS G13 item) refers to impaired will to move, and an altered timing of awareness of movement [66]. It belongs, along with G5 and several other items, to the disorganization factor [44]. In our female patients with schizophrenia, no associations between *COMT* genotypes or haplotypes and the severity of any PANSS sub-scores or individual items were observed. In line, no associations were found between PANSS item G13 and *COMT* rs4680 polymorphism in females, although female patients in that study had several other correlations [55]. In that cited study, females were about 19 years younger than our female patients, and had youth-onset schizophrenia [55]. However, when symptoms in the present study were categorized according to their severity, GG haplotype was less frequent in females with severe disturbance of volition, while the CG haplotype was over-represented in the group with severe symptoms, compared to those with mild symptoms. Disturbance of volition is commonly observed in individuals with schizophrenia. For example, it was the fifth-most-common PANSS single symptom in first-episode schizophrenia, reported in 65.4% of participants [67]. Moreover, the higher severity of disturbance of volition was, along with uncooperativeness, predictive of improved therapeutic benefit of paliperidone [68], but also during the initial asenapine treatment [69]. Despite being recognized as a common symptom in schizophrenia, disturbance of volition has received less attention than other PANSS items, and we were unable to find any data on the relationship between N5 item and either dopamine or *COMT* variants, but targeting such interactions appear as an attractive goal for future studies. 

Sex-specific correlations of *COMT* polymorphism with different parameters have been previously reported in both preclinical and clinical studies. In preclinical studies, the administration of COMT inhibitor tolcapone produced more prominent changes in dopamine metabolism in female than male rat brains [70], while ovariectomy decreased, and estradiol replacement restored COMT levels, respectively [71]. In healthy women, increases in estradiol during the menstrual cycle were associated with improved working memory performance for participants who were GG carriers, and an association was in the opposite direction for AA carriers [72]. Sex divergent effects of *COMT* rs4680 polymorphism have been shown previously on the effects on the relationship between executive dysfunction and psychotic-like experience in college students [22]. In addition, among healthy individuals with *COMT* rs4633-rs4680 T allele, males had higher COMT specific activity in serum than females [73]. Moreover, the higher number of associations in males in our study may also be related to their substantially higher severity as well as wider range of severities in PANSS total scores and all other scales in male patients. 

While the associations were sex-specific, in general 1) positive psychotic symptoms were not associated with any *COMT* genotypes or haplotype combinations, and 2) the presence of mainly high-activity *COMT* genotypes and haplotype was correlated with lower psychopathology on some indicators of PANSS negative and general psychopathology symptoms. Such findings somehow contribute to the general view that COMT is the most important regulator of dopamine function in PFC, but has a minor role in striatum and nucleus accumbens [74], whereas positive symptoms of schizophrenia are connected with hyperactivity of dopaminergic neurotransmission in striatum. However, an association between high-activity COMT and lower intensity of some negative and general symptoms is surprising, seeing that negative and cognitive symptoms are thought to arise from hypo-dopaminergic functioning in the frontal lobe and additional mesolimbic structures [75,76]. However, such findings were not detected in all patients with schizophrenia, and, given the high variance in dopaminergic measures in this population [77], it remains to be determined what is the role of *COMT* polymorphism in the complex landscape of symptoms in schizophrenia. 

In addition, polygenic risk score (PGS) was associated with the genetic liability for schizophrenia, but it is not yet applied in precision medicine at the individual level. For example, it predicted treatment resistance in one study [78], but was not associated with the onset of treatment-resistance in another study [79]. Moreover, PGS predicted lower antipsychotic response in first-episode patients [80], but was not associated with the variance in cognitive test scores [81] or poorer cognitive outcome in schizophrenia [82]. Maybe the biggest obstacle for establishing precision medicine in schizophrenia is the lack of laboratory tools to predict the outcome. Moreover, schizophrenia is not a single disorder, but rather a group of distinct disorders with some overlapping symptoms [83]. Our correlations of *COMT* variants, implicated in higher dopamine degradation with lower severity of particular symptoms, may be an interesting topic for future longitudinal studies. Such studies should address the predictive value of different *COMT* genotypes and haplotypes on the trajectories of symptoms. Of note, genetics itself will probably never be a single factor that determines the severity of the outcome of schizophrenia, due to the presence of many other factors unrelated to genes, such as non-compliance, stressful life events, and substance abuse. Nevertheless, we hypothesize that different *COMT* variants may contribute to our understanding on the complexity of psychopathology in schizophrenia and our results may provide a small contribution to the field. 

Taking into consideration that all our associations were sex-specific, we also recommend separate analyses for males and females in schizophrenia, or controlling the results for patient’s sex, especially in relation with studies addressing *COMT* variants. Our results add to the current knowledge on sexually dimorphic influence of *COMT* gene upon psychiatric phenotypes [84], that was also reported in healthy individuals [85]. In addition, it would be interesting to explore how different *COMT* genotypes and their haplotype combinations relate to psychopathology in different stages throughout the course of schizophrenia in male and female participants. 

It is intriguing to put the *COMT* gene into evolutionary perspective. Due to its influence in the prefrontal cortex in complex cognitive functioning, and the role of COMT enzyme in modulating prefrontal dopamine levels, *COMT* functional polymorphisms may have impacted adaptation to new environments during human evolution. Of note, G allele of *COMT* rs4680 is the ancestral allele, while A allele is a result of mutation, which is considered advantageous for human evolution because A allele was unique to humans and related to higher intelligence [86] and better insight problem solving [85]. Schizophrenia is very rare in societies with hunter-gatherer lifestyles [83]. In turn, hunter-gatherers had lower frequency of A allele than farming or industrialized populations [86]. While it is unknown if COMT gene has any role in the evolutionary paradox in schizophrenia, we speculate that the relation between COMT polymorphisms and haplotype combinations (associated with higher enzymatic activity), with lower intensity of some symptoms in schizophrenia, may suggest distinct impact of the *COMT* genes on schizophrenia onset and schizophrenia severity. 

### Limitations and Strengths

This study was cross-sectional, and therefore the observed associations do not imply a causal relationship. The severity of several PANSS items/symptoms was associated with *COMT* genotypes and haplotypes in patients with schizophrenia.

Limitation is that only two polymorphisms were determined. However, it has been shown that *COMT* rs4680 is a functional polymorphism affecting COMT activity, protein abundance, and protein stability [17], while rs4680 and rs4818, but not rs737865 or rs165599 genotypes, were associated with altered levels of S-COMT, in the human dorsolateral prefrontal cortex (DLPFC) [59]. In a preclinical study, COMT knockout mice had 60% higher dopamine levels in PFC, but not in nucleus accumbens [87]. However, in a postmortem study, the mean abundance of COMT mRNA in DLPFC of psychiatric patients, including those with schizophrenia, did not differ by the *COMT* genotype [88]. Therefore, the relationship between *COMT* genetic variants and COMT activity in brain cortical areas is complex and incompletely understood. Many other factors may also influence COMT activity, such as *COMT* methylation level. For example, participants with schizophrenia who received atypical antipsychotics, most often risperidone, had lower incidence of *COMT* DNA methylation than patients on typical antipsychotics [65]. 

However, this study has strengths, such as large sample size and the ethnically homogenous Caucasian population, which is important given the pronounced differences in prevalence of *COMT* genotypes and haplotypes across different ethnic groups. Apart from genotype frequencies, psychopathology ratings may also differ across different ethnic groups. For example, patients from China scored 20–30% higher on 5 PANSS items, including disturbance of volition, whereas they had 20% lower scores on another 11 items, including abstract thinking, compared to American inpatients [89]. Moreover, while the vast majority of studies on *COMT* investigated the rs4680 genotype, the present study included *COMT* rs4680 and rs4818 polymorphisms, which is important because it was suggested that the *COMT* rs4818 polymorphism may be responsible for an even larger variation in the COMT activity than the rs4680 polymorphism [20], and that *COMT* haplotypes are associated with schizophrenia susceptibility [90]. In addition, the frequency of the *COMT* rs4680 and rs4818 genetic variants was controlled for the effect of smoking and sex.

## 5. Conclusions

Our findings demonstrated sex-specific associations of *COMT high-activity* rs4680 and rs4818 variants and haplotype combinations with mostly lower severity of several individual PANSS symptoms in patients with schizophrenia. Longitudinal studies are needed to establish whether those correlations remain significant during treatment and if different *COMT* genotypes and haplotypes have predictive value on trajectories of treatment. 

## Figures and Tables

**Figure 1 genes-14-01358-f001:**
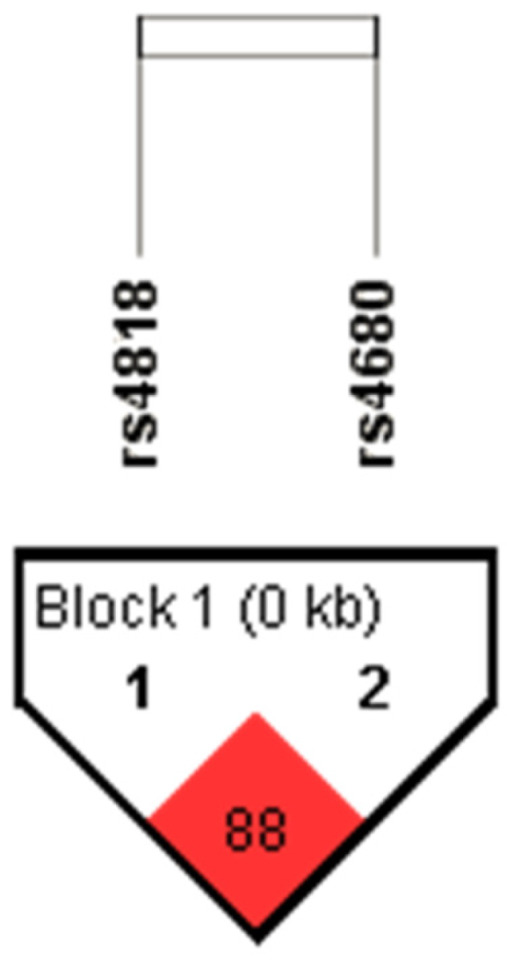
The LD plot of the rs4818 and rs4680 polymorphisms located in the *COMT* gene. Pairwise LD value (×100), as denoted in a bright red rectangle (D’ = 88), indicates a strong link between *COMT* rs4818 and rs4680 polymorphisms.

**Figure 2 genes-14-01358-f002:**
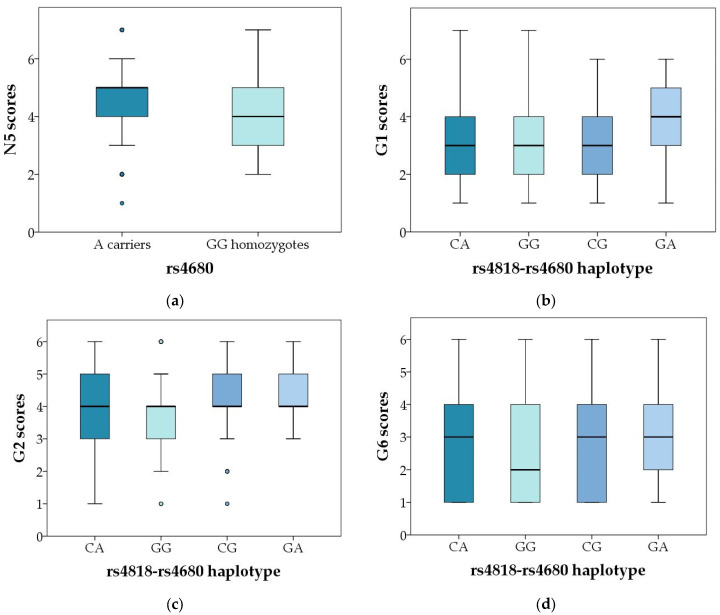
Significant genotypic and haplotypic association of *COMT* rs4818 and rs4680 polymorphisms with particular symptoms of schizophrenia. (**a**) Scores in the PANSS N5 item in male carriers of different *COMT* rs4680 genotypes and alleles, *p* = 0.023; (**b**) PANSS G1 scores in male carriers of different *COMT* rs4818–4680 haplotypes; *p* = 0.011; (**c**) PANSS G2 scores in male carriers of different *COMT* rs4818–4680 haplotypes; *p* = 0.003; and (**d**) PANSS G6 scores in male carriers of different *COMT* rs4818–4680 haplotypes; *p* = 0.020, in the male carriers of different *COMT* rs4818–4680 haplotypes. Central box represents the interquartile range, the middle line represents the median, the vertical line extends from the minimum to the maximum value, while separate dots represent the outliers.

**Figure 3 genes-14-01358-f003:**
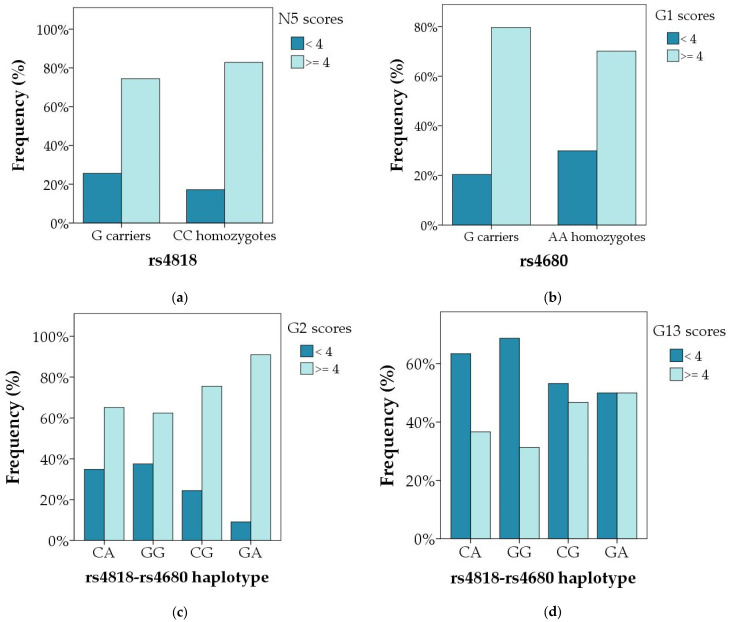
Significant differences in the distribution of the genotypes and haplotypes of the *COMT* rs4818 and rs4680 polymorphisms between subjects with mild and severe particular symptoms of schizophrenia (**a**) N5 item in male carriers of different *COMT* rs4818 genotypes and alleles; *p* = 0.022; (**b**) G1 item in male carriers of different *COMT* rs4680 genotypes and alleles; *p* = 0.024; (**c**) G2 item in male carriers of different *COMT* rs4818–4680 haplotypes; *p* = 0.003; and (**d**) G13 item in female carriers of different *COMT* rs4818–4680 haplotypes; *p* = 0.025. Data are presented as percentage of patients with mild (score < 4) and severe (score ≥ 4) symptoms in each genotypic and haplotypic group.

**Table 1 genes-14-01358-t001:** Demographic data of enrolled schizophrenia patients.

	Male Subjects	Female Subjects	Statistics
Age	38 (30; 47)	48 (38; 56)	U = 28,290.0; ***p* < 0.001**
Smoking (Yes/No)	376 (69.1%)/168 (30.9%)	199 (51.7%)/186 (48.3%)	χ^2^ = 29.038; ***p* < 0.001**
PANSS total	116 (98; 133)	93 (78; 104)	U = 50,937.5; ***p* < 0.001**
PANSS positive	30 (24; 36)	23 (18; 27)	U = 52,917.0; ***p* < 0.001**
PANSS negative	29 (24; 35)	23 (20; 26)	U = 55,396.5; ***p* < 0.001**
PANSS general	56 (49; 63)	46 (39; 52)	U = 57,075.0; ***p* < 0.001**
PANSS cognitive	15 (12; 18)	12 (10; 13)	U = 55,604.0; ***p* < 0.001**

**Table 2 genes-14-01358-t002:** The distribution of the genotypes and alleles of the *COMT* rs4818 and rs4680 polymorphisms, and *COMT* rs4818–rs4680 haplotype block in patients with schizophrenia divided by sex and smoking status.

SNP		Male Subjects	Female Subjects	Statistics	Smokers	Non-Smokers	Statistics
*COMT* rs4818	CC	204 (37.5%)	148 (38.4%)	χ^2^ = 0.560; df = 2; *p* = 0.756	228 (39.7%)	124 (35.0%)	χ^2^ = 2.183; df = 2; *p* = 0.336
	CG	261 (48.0%)	176 (45.7%)		265 (46.1%)	172 (48.6%)	
	GG	79 (14.5%)	61 (15.8%)		82 (14.3%)	58 (16.4%)	
	C	465 (85.5%)	324 (84.2%)	χ^2^ = 0.308; df = 1; *p* = 0.579	493 (85.7%)	296 (83.6%)	χ^2^ = 0.772; df = 1; *p* = 0.380
	GG	79 (14.5%)	61 (15.8%)		82 (14.3%)	58 (16.4%)	
	G	340 (62.5%)	237 (61.6%)	χ^2^ = 0.085; df = 1; *p* = 0.771	347 (60.3%)	230 (65.0%)	χ^2^ = 1.991; df = 1; *p* = 0.158
	CC	204 (37.5%)	148 (38.4%)		228 (39.7%)	124 (35.0%)	
	C	669 (61.5%)	472 (61.3%)	χ^2^ = 0.007; df = 1; *p* = 0.934	721 (62.7%)	420 (59.3%)	χ^2^ = 2.105; df = 1; *p* = 0.147
	G	419 (38.5%)	298 (38.7%)		429 (37.3%)	288 (40.7%)	
*COMT* rs4680	AA	143 (26.3%)	95 (24.7%)	χ^2^ = 0.919; df = 2; *p* = 0.632	154 (26.8%)	84 (23.7%)	χ^2^ = 1.140; df = 2; *p* = 0.566
	GA	270 (49.6%)	187 (48.6%)		280 (48.7%)	177 (50.0%)	
	GG	131 (24.1%)	103 (26.8%)		141 (24.5%)	93 (26.3%)	
	A	413 (75.9%)	282 (73.2%)	χ^2^ = 0.854; df = 1; *p* = 0.355	434 (75.5%)	261 (73.7%)	χ^2^ = 0.356; df = 1; *p* = 0.551
	GG	131 (24.1%)	103 (26.8%)		141 (24.5%)	93 (26.3%)	
	G	417 (76.7%)	298 (77.4%)	χ^2^ = 0.071; df = 1; *p* = 0.790	438 (76.2%)	277 (78.2%)	χ^2^ = 0.532; df = 1; *p* = 0.466
	AA	127 (23.3%)	87 (22.6%)		137 (23.8%)	77 (21.8%)	
	A	556 (51.1%)	377 (49.0%)	χ^2^ = 0.827; df = 1; *p* = 0.363	588 (51.1%)	345 (48.7%)	χ^2^ = 1.011; df = 1; *p* = 0.315
	G	532 (48.9%)	393 (51.0%)		562 (48.9%)	363 (51.3%)	
*COMT* rs4818–rs4680 haplotype	CA	534 (49.1%)	363 (47.1%)	χ^2^ = 1.515; df = 3; *p* = 0.679	564 (49.0%)	333 (47.0%)	χ^2^ = 3.030; df = 3; *p* = 0.387
	GG	397 (36.5%)	284 (36.9%)		405 (35.2%)	276 (39.0%)	
	CG	135 (12.4%)	109 (14.2%)		157 (13.7%)	87 (12.3%)	
	GA	22 (2.0%)	14 (1.8%)		24 (2.1%)	12 (1.7%)	

**Table 3 genes-14-01358-t003:** The scores on PANSS total, positive, negative, general psychopathology and cognitive subscales, as well as on particular PANSS items, in male subjects with schizophrenia divided by carriers of different genotypes, alleles and haplotypes of the *COMT* rs4818 and rs4680 polymorphisms.

PANSS Scores	*COMT* rs4818				*COMT* rs4680				rs4818–rs4680 Haplotype
	CC vs. GC vs. GG	C vs. GG	G vs. CC	C vs. G	AA vs. GA vs. GG	A vs. GG	G vs. AA	A vs. G	
PANSS total	0.850	0.757	0.724	0.928	0.711	0.565	0.640	0.902	0.794
PANSS positive	0.929	0.722	0.982	0.868	0.581	0.993	0.371	0.536	0.904
P1	0.803	0.983	0.526	0.648	0.833	0.889	0.785	0.647	0.928
P2	0.770	0.642	0.707	0.979	0.247	0.505	0.260	0.722	0.967
P3	0.675	0.404	0.582	0.415	0.652	0.611	0.691	0.969	0.674
P4	0.618	0.331	0.658	0.419	0.223	0.957	0.088	0.289	0.769
P5	0.687	0.799	0.387	0.460	0.525	0.904	0.317	0.447	0.854
P6	0.667	0.435	0.861	0.782	0.958	0.965	0.625	0.893	0.740
P7	0.507	0.244	0.724	0.398	0.260	0.407	0.165	0.132	0.400
PANSS negative	0.688	0.840	0.389	0.478	0.391	0.190	0.881	0.405	0.869
N1	0.602	0.914	0.325	0.455	0.227	0.100	0.910	0.297	0.740
N2	0.642	0.565	0.602	0.943	0.745	0.478	0.720	0.454	0.692
N3	0.404	0.553	0.339	0.712	0.934	0.762	0.651	0.711	0.973
N4	0.616	0.503	0.370	0.330	0.324	0.134	0.849	0.204	0.328
N5	0.272	0.210	0.173	0.109	0.060	**0.023**	0.695	0.141	0.401
N6	0.871	0.755	0.618	0.609	0.581	0.416	0.951	0.780	0.719
N7	0.802	0.554	0.923	0.814	0.223	0.191	0.532	0.690	0.967
PANSS general	0.753	0.543	0.821	0.879	0.941	0.784	0.594	0.924	0.164
G1	0.577	0.397	0.752	0.833	0.468	0.690	0.809	0.313	**0.011**
G2	0.254	0.939	0.111	0.246	0.569	0.887	0.781	0.616	**0.003**
G3	0.839	0.693	0.771	0.998	0.952	0.999	0.587	0.853	0.038
G4	0.580	0.380	0.415	0.306	0.784	0.608	0.236	0.487	0.126
G5	0.770	0.488	0.976	0.706	0.554	0.277	0.643	0.373	0.814
G6	0.214	0.536	0.079	0.121	0.341	0.870	0.369	0.323	**0.020**
G7	0.336	0.760	0.203	0.460	0.188	0.476	0.339	0.683	0.297
G8	0.161	0.089	0.172	0.067	0.262	0.485	0.104	0.151	0.199
G9	0.789	0.570	0.583	0.498	0.663	0.436	0.856	0.719	0.900
G10	0.197	0.336	0.080	0.085	0.222	0.844	0.047	0.241	0.239
G11	0.541	0.361	0.760	0.800	0.042	0.084	0.293	0.735	0.990
G12	0.888	0.660	0.735	0.643	0.090	0.282	0.156	0.807	0.798
G13	0.969	0.832	0.848	0.808	0.977	0.858	0.767	0.827	0.935
G14	0.272	0.114	0.414	0.166	0.240	0.386	0.137	0.119	0.450
G15	0.584	0.314	0.931	0.564	0.747	0.476	0.966	0.673	0.607
G16	0.296	0.721	0.120	0.201	0.253	0.455	0.125	0.140	0.309
PANSS cognitive	0.916	0.782	0.835	0.996	0.105	0.252	0.217	0.908	0.993

The data are represented as *p*-values obtained with Kruskal–Wallis ANOVA or Mann–Whitney test; PANSS = Positive and Negative Syndrome Scale; PANSS general = PANSS general psychopathology scores.

**Table 4 genes-14-01358-t004:** The scores on PANSS total, positive, negative, general psychopathology, and cognitive subscales, as well as on particular PANSS items, in female subjects with schizophrenia divided by carriers of different genotypes, alleles and haplotypes of the *COMT* rs4818 and rs4680 polymorphisms.

PANSS Scores	*COMT* rs4818				*COMT* rs4680				rs4818–rs4680 Haplotype
	CC vs. GC vs. GG	C vs. GG	G vs. CC	C vs. G	AA vs. GA vs. GG	A vs. GG	G vs. AA	A vs. G	
PANSS total	0.817	0.988	0.546	0.664	0.455	0.386	0.955	0.836	0.199
PANSS positive	0.926	0.744	0.925	0.915	0.478	0.421	0.835	0.878	0.249
P1	0.915	0.839	0.781	0.929	0.948	0.749	0.551	0.820	0.542
P2	0.603	0.907	0.369	0.567	0.882	0.824	0.897	0.657	0.411
P3	0.499	0.247	0.852	0.456	0.150	0.101	0.800	0.439	0.638
P4	0.735	0.437	0.864	0.593	0.909	0.744	0.456	0.914	0.294
P5	0.928	0.741	0.764	0.698	0.387	0.970	0.648	0.449	0.163
P6	0.774	0.777	0.475	0.513	0.109	0.253	0.381	0.956	0.399
P7	0.754	0.453	0.797	0.562	0.576	0.297	0.286	0.420	0.061
PANSS negative	0.815	0.772	0.526	0.547	0.680	0.554	0.881	0.902	0.154
N1	0.744	0.555	0.506	0.434	0.478	0.327	0.562	0.218	0.614
N2	0.880	0.765	0.779	0.968	0.421	0.242	0.922	0.524	0.498
N3	0.876	0.733	0.632	0.603	0.740	0.854	0.804	0.558	0.294
N4	0.310	0.808	0.133	0.234	0.527	0.464	0.839	0.905	0.095
N5	0.719	0.452	0.974	0.707	0.302	0.353	0.810	0.944	0.604
N6	0.680	0.421	0.545	0.393	0.378	0.259	0.975	0.635	0.445
N7	0.445	0.296	0.299	0.198	0.780	0.608	0.899	0.880	0.103
PANSS general	0.708	0.953	0.424	0.551	0.564	0.435	0.948	0.814	0.271
G1	0.866	0.881	0.593	0.647	0.135	0.338	0.160	0.070	0.185
G2	0.765	0.558	0.540	0.457	0.200	0.367	0.492	0.895	0.572
G3	0.495	0.734	0.236	0.309	0.889	0.627	0.921	0.678	0.715
G4	0.483	0.570	0.421	0.790	0.624	0.332	0.415	0.410	0.648
G5	0.453	0.214	0.545	0.277	0.892	0.770	0.835	0.644	0.146
G6	0.365	0.158	0.533	0.234	0.524	0.262	0.723	0.298	0.596
G7	0.490	0.678	0.363	0.672	0.522	0.706	0.367	0.764	0.545
G8	0.998	0.979	0.960	0.983	0.433	0.796	0.412	0.610	0.806
G9	0.835	0.631	0.614	0.541	0.654	0.718	0.787	0.855	0.547
G10	0.192	0.084	0.263	0.088	0.644	0.353	0.601	0.394	0.321
G11	0.321	0.300	0.502	0.939	0.593	0.326	0.422	0.326	0.244
G12	0.899	0.853	0.739	0.891	0.995	0.959	0.699	0.995	0.555
G13	0.138	0.535	0.047	0.083	0.938	0.994	0.740	0.842	0.062
G14	0.978	0.861	0.867	0.832	0.973	0.827	0.375	0.815	0.585
G15	0.071	0.206	0.170	0.764	0.092	0.204	0.337	0.963	0.888
G16	0.608	0.993	0.347	0.504	0.275	0.959	0.434	0.381	0.221
PANSS cognitive	0.753	0.841	0.453	0.524	0.856	0.578	0.495	0.619	0.221

The data are represented as *p*-values obtained with Kruskal–Wallis ANOVA or Mann–Whitney test; PANSS = Positive and Negative Syndrome Scale; PANSS general = PANSS general psychopathology scores.

**Table 5 genes-14-01358-t005:** The distribution of the genotypes and alleles of the *COMT* rs4818 and rs4680 polymorphisms, and their haplotype block, in male subjects with mild and severe symptoms of schizophrenia divided by scores on PANSS total, positive, negative, general psychopathology and cognitive subscales as well as on particular PANSS items.

PANSS Scores	*COMT* rs4818				*COMT* rs4680				rs4818–rs4680 Haplotype
	CC vs. GC vs. GG	C vs. GG	G vs. CC	C vs. G	AA vs. GA vs. GG	A vs. GG	G vs. AA	A vs. G	
PANSS total	0.685	0.423	0.563	0.414	0.675	0.803	0.485	0.767	0.831
PANSS positive	0.873	0.822	0.604	0.631	0.741	0.803	0.619	0.527	0.504
P1	0.741	0.911	0.490	0.669	0.852	0.571	0.668	0.644	0.579
P2	0.747	0.788	0.445	0.500	0.328	0.523	0.378	0.785	0.607
P3	0.873	0.611	0.787	0.652	0.362	0.686	0.338	0.258	0.177
P4	0.667	0.368	0.757	0.497	0.690	0.514	0.323	0.389	0.842
P5	0.952	0.814	0.787	0.756	0.704	0.686	0.452	0.447	0.861
P6	0.763	0.470	0.723	0.535	0.853	0.765	0.366	0.600	0.773
P7	0.483	0.261	0.439	0.263	0.201	0.176	0.147	0.073	0.298
PANSS negative	0.539	0.893	0.277	0.404	0.237	0.128	0.935	0.426	0.853
N1	0.479	0.884	0.235	0.363	0.611	0.486	0.985	0.878	0.587
N2	0.530	0.504	0.518	0.911	0.842	0.618	0.905	0.821	0.980
N3	0.317	0.509	0.279	0.672	0.970	0.927	0.998	0.835	0.825
N4	0.913	0.747	0.714	0.672	0.883	0.627	0.938	0.643	0.630
N5	0.064	0.211	**0.022**	**0.025**	0.117	0.038	0.183	0.089	0.076
N6	0.972	0.863	0.835	0.814	0.591	0.993	0.601	0.552	0.669
N7	0.719	0.476	0.887	0.791	0.295	0.273	0.463	0.811	0.961
PANSS general	0.777	0.906	0.531	0.704	0.410	0.366	0.325	0.190	0.181
G1	0.094	0.046	0.146	0.041	0.280	0.386	**0.024**	0.136	0.054
G2	0.382	0.771	0.168	0.263	0.532	0.589	0.968	0.890	**0.003**
G3	0.974	0.873	0.837	0.821	0.798	0.916	0.938	0.635	0.027
G4	0.677	0.832	0.379	0.467	0.253	0.744	0.027	0.223	0.187
G5	0.191	0.136	0.606	0.690	0.332	0.198	0.999	0.538	0.882
G6	0.398	0.802	0.236	0.481	0.878	0.929	0.907	0.824	0.096
G7	0.541	0.289	0.978	0.574	0.418	0.415	0.795	0.956	0.534
G8	0.284	0.245	0.163	0.115	0.310	0.509	0.176	0.178	0.388
G9	0.901	0.672	0.980	0.842	0.505	0.992	0.345	0.499	0.585
G10	0.475	0.346	0.665	0.859	0.832	0.548	0.477	0.660	0.044
G11	0.281	0.281	0.442	0.991	0.067	0.125	0.291	0.823	0.985
G12	0.774	0.517	0.621	0.497	0.108	0.102	0.492	0.583	0.780
G13	0.915	0.765	0.707	0.676	0.892	0.953	0.636	0.819	0.930
G14	0.362	0.156	0.565	0.258	0.431	0.473	0.139	0.223	0.453
G15	0.259	0.128	0.281	0.124	0.498	0.923	0.089	0.541	0.132
G16	0.203	0.546	0.162	0.499	0.639	0.834	0.558	0.711	0.555
PANSS cognitive	0.882	0.679	0.688	0.531	0.670	0.471	0.789	0.449	0.807

The data are represented as *p*-values obtained with χ^2^-test; PANSS = Positive and Negative Syndrome Scale; PANSS general = PANSS general psychopathology scores.

**Table 6 genes-14-01358-t006:** The distribution of the genotypes and alleles of the *COMT* rs4818 and rs4680 polymorphisms, and their haplotype block, in female subjects with mild and severe symptoms of schizophrenia divided by scores on PANSS total, positive, negative, general, and cognitive subscales as well as on particular PANSS items.

PANSS Scores	*COMT* rs4818				*COMT* rs4680				rs4818–rs4680 Haplotype
	CC vs. GC vs. GG	C vs. GG	G vs. CC	C vs. G	AA vs. GA vs. GG	A vs. GG	G vs. AA	A vs. G	
PANSS total	0.276	0.344	0.123	0.112	0.032	0.403	0.071	0.034	0.098
PANSS positive	0.309	0.189	0.770	0.625	0.709	0.451	0.928	0.407	0.791
P1	0.526	0.543	0.491	0.870	0.765	0.702	0.989	0.971	0.334
P2	0.403	0.879	0.190	0.314	0.855	0.590	0.756	0.713	0.517
P3	0.807	0.566	0.921	0.815	0.865	0.601	0.700	0.717	0.986
P4	0.843	0.802	0.681	0.875	0.860	0.969	0.322	0.779	0.354
P5	0.825	0.546	0.940	0.709	0.403	0.640	0.706	0.740	0.136
P6	0.904	0.660	0.815	0.690	0.218	0.303	0.512	0.956	0.285
P7	0.932	0.722	0.814	0.723	0.393	0.172	0.398	0.254	0.319
PANSS negative	0.947	0.776	0.797	0.740	0.264	0.925	0.371	0.393	0.758
N1	0.768	0.575	0.531	0.460	0.468	0.256	0.633	0.220	0.645
N2	0.731	0.453	0.980	0.678	0.322	0.134	0.660	0.239	0.378
N3	0.998	0.994	0.955	0.972	0.877	0.774	0.847	0.998	0.156
N4	0.787	0.932	0.500	0.602	0.929	0.712	0.537	0.710	0.243
N5	0.603	0.802	0.407	0.650	0.075	0.071	0.987	0.476	0.551
N6	0.668	0.400	0.560	0.391	0.434	0.268	0.862	0.586	0.281
N7	0.604	0.352	0.500	0.332	0.671	0.404	0.710	0.605	0.049
PANSS general	0.028	0.083	**0.012**	**0.007**	**0.005**	**0.006**	0.045	**0.001**	**0.007**
G1	0.645	0.740	0.479	0.746	0.381	0.543	0.128	0.221	0.517
G2	0.864	0.597	0.943	0.741	0.411	0.501	0.586	0.924	0.942
G3	0.813	0.909	0.578	0.740	0.669	0.925	0.619	0.556	0.424
G4	0.605	0.414	0.790	0.807	0.878	0.685	0.563	0.606	0.727
G5	0.570	0.291	0.660	0.384	0.766	0.678	0.740	0.997	0.093
G6	0.587	0.730	0.302	0.362	0.959	0.995	0.821	0.864	0.533
G7	0.321	0.734	0.135	0.216	0.829	0.890	0.279	0.647	0.600
G8	0.861	0.586	0.813	0.648	0.347	0.582	0.453	0.757	0.871
G9	0.503	0.979	0.267	0.425	0.463	0.953	0.410	0.509	0.296
G10	0.783	0.485	0.799	0.582	0.652	0.423	0.961	0.685	0.599
G11	0.422	0.428	0.476	0.934	0.943	0.831	0.680	0.979	0.517
G12	0.454	0.246	0.955	0.565	0.625	0.446	0.847	0.768	0.462
G13	0.131	0.356	0.046	0.057	0.912	0.728	0.730	0.663	**0.025**
G14	0.878	0.623	0.765	0.637	0.884	0.839	0.790	0.929	0.638
G15	0.121	0.171	0.333	0.966	0.141	0.111	0.504	0.505	0.834
G16	0.283	0.964	0.140	0.308	0.221	0.873	0.345	0.261	0.049
PANSS cognitive	0.296	0.284	0.152	0.635	0.349	0.574	0.106	0.862	0.700

The data are represented as *p*-values obtained with χ^2^-test; PANSS = Positive and Negative Syndrome Scale; PANSS general = PANSS general psychopathology scores.

## Data Availability

The data presented in this study are available on request.

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
