# Peer review of "Genotypic and Haplotypic Association of Catechol-O-Methyltransferase rs4680 and rs4818 Gene Polymorphisms with Particular Clinical Symptoms in Schizophrenia"

_genes, 2023, doi:10.3390/genes14071358_

Round 1
Reviewer 1 Report
Dear Author
The target gene is catechol-O-methyl- 1 transferase and two SNPs are selected. The "Introduction" is too long. However, I can not find the molecular role of catechol-O-methyl- 1 transferase. I expect you indicate its methylation role and gene downstream of the regulation. Does it have any role in other gene promoter methylation? Where is the exact locus of SNPs in the 3D structure of the catechol-O-methyl- 1 transferase? binding site or catalytic site? Why select these two SNPs?
A long Introduction is not recommended in a high-quality manuscript. Most parts can transfer to the discussion.
There is no informed consent so how do you solve the ethical considerations, how do you select patients?
In the method, you just indicated to the ABI and Taqman? If you used real-time PCR what is the thermocycler program? what is the sequence of primers to detect rs4680 and rs4818 polymorphisms?
In Figure 22 just G1, G2, and G6 scores are reported and it is not clear why?
Several published articles have already indicated the importance of rs4818 and rs4680 polymorphisms in schizophrenia, so what is the novelty of your work?
Subtitling of discussion is strange to me. If the journal format accepts this I think it would be OK.
1- Nikolac Perkovic M, Sagud M, Zivkovic M, Uzun S, Nedic Erjavec G, Kozumplik O, Svob Strac D, Mimica N, Mihaljevic Peles A, Pivac N. Catechol-O-methyltransferase rs4680 and rs4818 haplotype association with treatment response to olanzapine in patients with schizophrenia. Scientific Reports. 2020 Jun 22;10(1):10049.
2-Sagud M, Tudor L, Uzun S, Perkovic MN, Zivkovic M, Konjevod M, Kozumplik O, Vuksan Cusa B, Svob Strac D, Rados I, Mimica N. Haplotypic and genotypic association of catechol-O-methyltransferase rs4680 and rs4818 polymorphisms and treatment resistance in schizophrenia. Frontiers in Pharmacology. 2018 Jul 3;9:705.
and more.....
Moderate editing of English language required
Author Response
Reviewer No 1.
The target gene is catechol-O-methyl- 1 transferase and two SNPs are selected. The "Introduction" is too long. However, I can not find the molecular role of catechol-O-methyl- 1 transferase. I expect you indicate its methylation role and gene downstream of the regulation. Does it have any role in other gene promoter methylation?
Answer: We have accepted this comment. Thank you very much for raising this important issue. Catechol-O-methyltransferase is an enzyme involved in the inactivation of compounds having a catechol structure, including catecholamine neurotransmitters, by introducing a methyl group, donated by S-adenosyl methionine, to the catecholamine. Thus, we are not familiar with its role in regulation of other genes' expression by methylation. To improve our manuscript according to reviewer’s comment, we included the following text in our manuscript: “COMT is an enzyme involved in the inactivation of compounds having a catechol structure, including catecholamine neurotransmitters such as dopamine, by introducing a methyl group, donated by S-adenosyl methionine, to the catecholamine. Byproducts of this reaction are O-methylated catechol and S-adenosyl-L-homocysteine.”
Where is the exact locus of SNPs in the 3D structure of the catechol-O-methyl- 1 transferase? binding site or catalytic site?
Answer: We have accepted this comment and added in the text “The active site of COMT consists of the S-adenosyl-L-methionine (SAM) binding domain and the catalytic site. The catalytic site contains a metal ion (Mg2+) and amino acids important for substrate binding and catalysis of the methylation reaction. The polymorphic residue according to rs4680 is buried in a hydrophobic residue, around 16AËš away from the SAM-binding site (Jatana et al., 2013), and in a complementary hydrophobic methyl binding pocket according to rs4818 (Zubieta et al., 2002). “
References 14 and 15 were deleted from the text, and replaced by Jatana et al, 2013. (14) and Zubieta et al, (15).
- Jatana, N.; Sharma, A.; Latha, N. Pharmacophore modeling and virtual screening studies to design potential COMT inhibitors as new leads. Mol. Graph. Model. 2013, 39, 145-164. doi: 10.1016/j.jmgm.2012.10.010.
- Zubieta, C.; Kota, P.; Ferrer, J.L.; Dixon, R.A.; Noel, J.P. Structural basis for the modulation of lignin monomer methylation by caffeic acid/5-hydroxyferulic acid 3/5-O-methyltransferase. Plant Cell. 2002,14(6), 1265-1277. doi: 10.1105/tpc.001412.
Why select these two SNPs?
Answer: We have accepted this comment. We have selected these two SNPs because these two functional polymorphisms substantially affect COMT activity and therefore influence dopamine levels in prefrontal cortex. This is highlighted in the Introduction: “Specifically, COMT rs4680 polymorphism, or a G/A substitution, leads to Valine (Val) replacement with methionine (Met) at codon 158 of membrane bound COMT (MB-COMT) and at codon 108 of soluble short form (S-COMT) [18], and results in significant (three- to four-fold) fall of the COMT activity in the A (Met) carriers. Another COMT polymorphism, COMT rs4818 polymorphism is located on exon 4, and consist of a C/G substitution (Leu/Leu) at codon 86 of the S-COMT and at codon 136 of the MB-COMT [19]. Since GG carriers have higher COMT activity than the CC carriers of this polymorphism, presence of the G variant is related to greater COMT activity and reduced prefrontal dopamine activity [19]. It is assumed that COMT activity is more under influence of the COMT rs4818 than COMT rs4680 polymorphism [20].”
However, there are many other COMT polymorphisms, and that was mentioned in the ”Limitations”.
It was also mentioned in limitations: Limitation is that only two polymorphisms were determined. However, it has been shown that COMT rs4680 is a functional polymorphism affecting COMT activity, protein abundance, and protein stability [17], while rs4680 and rs4818, but not rs737865 or rs165599 genotypes, were associated with altered levels of S-COMT, in the human dorsolateral prefrontal cortex (DLPFC) [59].
A long Introduction is not recommended in a high-quality manuscript. Most parts can transfer to the discussion.
Answer: We have accepted this comment, and have reduced the text in the Introduction section. With the above sections added in the Introduction, it has now 745 words, while in the initial version the Introduction had 1050 words.
There is no informed consent so how do you solve the ethical considerations, how do you select patients?
Answer: We have accepted this comment. In “Methods” section it was already mentioned that “Before participation, patients were informed in details about the procedures, and after they have signed the written informed consent, the study was carried out in accordance with The Code of Ethics of the World Medical Association (Declaration of Helsinki from 1975”).
In the method, you just indicated to the ABI and Taqman? If you used real-time PCR what is the thermocycler program? what is the sequence of primers to detect rs4680 and rs4818 polymorphisms?
Answer: We have accepted this comment. Genotyping procedures were performed using TaqMan genotyping commercial kits described by assay IDs. We cannot provide the primers’ sequences since probably this information is confidential and it is not mentioned in products’ data sheets. On the other hand, we have provided the thermal cycler conditions so we added the following text to ‘Materials and Methods’ section:
‘The COMT rs4818 (assay ID: C__2538750_10) and rs4680 (assay ID: C__25746809_50) polymorphisms genotypes were determined using the primers and probes from Applied Biosystems as TaqMan® Drug Metabolism Genotyping Assays (Applied Biosystems, Fos-ter City, CA, USA) on ABI Prism 7300 Real time PCR System apparatus (Applied Biosys-tems, Foster City, CA, USA), according to the procedures described by Applied Biosystems. Thermal cycler conditions were 10 min at 95°C followed by 50 cycles of denaturation at 92°C for 15 sec and elongation at 60°C for 90 sec.’
In Figure 22 just G1, G2, and G6 scores are reported and it is not clear why?
Answer: We have accepted this comment. Thank you for this comment. In the Figure 2, we have presented the scores that were statistically significant, while all others were not, and therefore, other nonsignificant scores were not reported in the Figure 2. Therefore, the p values were provided in the figure legend, as follows.
Figure 2. Significant genotypic and haplotypic association of COMT rs4818 and rs4680 polymor-phisms with particular symptoms of schizophrenia (2a) Scores in the PANSS N5 item in male carriers of different COMT rs4680 genotypes and alleles, p=0.023; (2b) PANSS G1 scores in male carriers of different COMT rs4818-4680 haplotypes; p=0.011; (2c) PANSS G2 scores in male carri-ers of different COMT rs4818-4680 haplotypes; p=0.003; and (2d) PANSS G6 scores in male carri-ers of different COMT rs4818-4680 haplotypes; p=0.020, in the male carriers of different COMT rs4818-4680 haplotypes. Central box represents the interquartile range, the middle line repre-sents the median, the vertical line extends from the minimum to the maximum value, while separate dots represent the outliers.
However, to accept your comment and to make our results more visible, we have added also a new figure, Figure 3, to present other significant results related to selected PANSS items and COMT genoptypic and haplotypic combinations. “Figure 3. Significant differences in the distribution of the genotypes and haplotypes of the COMT rs4818 and rs4680 polymorphisms between subjects with mild and severe particular symptoms of schizophrenia (3a) N5 item in male carriers of different COMT rs4818 genotypes and alleles; p=0.022; (3b) G1 item in male carriers of different COMT rs4680 genotypes and alleles; p=0.024; (3c) G2 item in male carriers of different COMT rs4818-4680 haplotypes; p=0.003; and (3d) G13 item in female carriers of different COMT rs4818-4680 haplotypes; p=0.025. Data are presented as percentage of patients with mild (score < 4) and severe (score >=4) symptoms in each genotypic and haplotypic group.
Several published articles have already indicated the importance of rs4818 and rs4680 polymorphisms in schizophrenia, so what is the novelty of your work?
Answer: Thank you for this thoughtful comment. The aim of the present article was not to investigate the association between COMT rs4818 and rs4680 polymorphisms in schizophrenia, but to determine the correlations between rs4818 and rs4680 polymorphisms and the resulting haplotype combination with each of the PANSS items and subscales. Given the role of dopamine turnover in particular symptoms of schizophrenia, we assumed that the genetics of dopamine-metabolizing enzyme COMT may be associated with specific symptoms.
Subtitling of discussion is strange to me. If the journal format accepts this I think it would be OK.
Answer: Thank you for this suggestion, we have accepted this comment and subtitles have been deleted.

Reviewer 2 Report
This study sought to assess genotypic and haplotypic associations between COMT gene polymorphisms and key clinical symptoms characteristic of schizophrenia.
Overall, the manuscript was well-written and clearly presented.
The introduction is thorough and well-referenced.
The statistical analyses are very well described and reported in detail in the results.
In the discussion, you mention certain future studies that could be carried out. I think your manuscript would be greatly strengthened by mentioned specifically the nature of such envisioned studies. You speak to future studies exploring the association of COMT polymorphisms and depressive symptoms in schizophrenia, and how they are linked to life stressors. You also mention the need for a number of longitudinal studies to check whether associations remain significant under treatment. This would be interesting to elaborate in a “Future directions” section of a few sentences to add (life historical) context to your study, in particular across time.
In addition, in the discussion, it would add clout to your work to discuss the clinical implications of your findings, e.g. in the context of polygenic risk scores, or in how the results add support for the need to clinically implement high throughput sequencing technologies more regularly. How do your results galvanize the field of (genetically guided) precision medicine?
Finally, if you deem it relevant, a quick discussion of the evolutionary advantage of these COMT variants – for example in the context of the evolutionary paradox of schizophrenia – would be fascinating in order to again frame the research within a broader, more philosophical molecular evolutionary framework.
Very minor edits to the English language can be made to improve the manuscript, throughout.
For example, line 72, “One of the most frequently investigated genes, associated with cognitive phenotypes, is a COMT gene” should read “One of the most frequently investigated genes, associated with cognitive phenotypes, is the COMT gene.”
As another example, line 258,
“Both COMT polymorphisms showed association with the PANSS general scores” should read “Both COMT polymorphisms showed an association with the PANSS general scores”.
25+ examples can be found throughout the text.
Author Response
This study sought to assess genotypic and haplotypic associations between COMT gene polymorphisms and key clinical symptoms characteristic of schizophrenia.
Overall, the manuscript was well-written and clearly presented.
The introduction is thorough and well-referenced.
Answer: Thank you. We have curtailed the “Introduction” according to the suggestions of the first reviewer.
The statistical analyses are very well described and reported in detail in the results.
Answer: Thank you so much
In the discussion, you mention certain future studies that could be carried out. I think your manuscript would be greatly strengthened by mentioned specifically the nature of such envisioned studies. You speak to future studies exploring the association of COMT polymorphisms and depressive symptoms in schizophrenia, and how they are linked to life stressors. You also mention the need for a number of longitudinal studies to check whether associations remain significant under treatment. This would be interesting to elaborate in a “Future directions” section of a few sentences to add (life historical) context to your study, in particular across time.
Answer: we have accepted this comment and have written in this part of Discussion:” Both G2 „anxiety “and G6 „depression “items are a part of PANSS anxiety and de-pression dimension, according to the factor analysis [62]. Our findings suggest that high COMT activity related to low dopamine availability was associated with the lowest anxiety and depression dimension scores in males. In another study in Croatian population, COMT rs4680 was not associated with affective/depressive PANSS factors [63]. Our results do not agree with the findings of the higher levels of depression in patients with COMTrs4680 GG genotype, compared to A carriers, in patients with first-episode schizophrenia, whereby COMT rs4680 polymorphism moderated an association between the severity of depression and stressful life events [64]. That study had small sample size, and in both cited studies males and females were not separately analyzed [63, 64]. Importantly, in patients with schizophrenia, COMT DNA methylation was inversely correlated with depressed subdomain of the PANSS, i.e., the higher the depressive symptoms, the lower the DNA methylation [65]. Future studies exploring the associations of COMT polymorphisms and depressive symptoms in schizophrenia need to consider childhood and life-time stressors, which may strongly influence this relationship.”
In addition, in the discussion, it would add clout to your work to discuss the clinical implications of your findings, e.g. in the context of polygenic risk scores, or in how the results add support for the need to clinically implement high throughput sequencing technologies more regularly. How do your results galvanize the field of (genetically guided) precision medicine?
Answer; We have accepted this comment and have added n the Discussion: “In addition, polygenic risk score(PGS) was associated with the genetic liability for schizophrenia, but it is not yet applied in precision medicine on the individual level. For example, it predicted treatment resistance in (Werner et al, 2020), but was not associated with the onset of treatment-resistance in another study (Wimberlay et al, 2017). Moreover, PGS predicted lower antipsychotic response in first-episode patients (Zhang et al, 2019), but was not associated with the variance in cognitive test scores (Richards et al, 2020) or poorer cognitive outcome in schizophrenia (Habtewold et al, 2020. Maybe the biggest obstacle for establishing precision medicine in schizophrenia is the lack of laboratory tools to predict the outcome. Moreover, schizophrenia is not a single disorder, but rather a group of distinct disorders with some overlapping symptoms (Rantala et al, 2022). Our correlations of COMT variants implicated in higher dopamine degradation with lower severity of some symptoms may be an interesting topic for future longitudinal studies. Such studies would address the predictive value of different COMT polymorphisms and haplotypes on the trajectories of symptoms. Of note, genetics itself will probably never be a single factor that determines the severity / outcome of schizophrenia, given the presence of many others unrelated to genes, such as non-compliance, stresfull life events and substance abuse. Nevertheless, we hypothesize that different COMT variants may contribute to our understanding on the complexity of psychopathology in schizophrenia and our results may provide a small contribution to the field. „
Wimberley T, Gasse C, Meier SM, Agerbo E, MacCabe JH, Horsdal HT. Polygenic Risk Score for Schizophrenia and Treatment-Resistant Schizophrenia. Schizophr Bull. 2017;43(5):1064-1069. doi: 10.1093/schbul/sbx007.
Zhang JP, Robinson D, Yu J, Gallego J, Fleischhacker WW, Kahn RS, Crespo-Facorro B, Vazquez-Bourgon J, Kane JM, Malhotra AK, Lencz T. Schizophrenia Polygenic Risk Score as a Predictor of Antipsychotic Efficacy in First-Episode Psychosis. Am J Psychiatry. 2019;176(1):21-28. doi: 10.1176/appi.ajp.2018.17121363.
Rantala MJ, Luoto S, Borráz-León JI, Krams I. Schizophrenia: The new etiological synthesis. Neurosci Biobehav Rev. 2022;142:104894. doi: 10.1016/j.neubiorev.2022.
Habtewold TD, Liemburg EJ, Islam MA, de Zwarte SMC, Boezen HM; GROUP Investigators; Bruggeman R, Alizadeh BZ. Association of schizophrenia polygenic risk score with data-driven cognitive subtypes: A six-year longitudinal study in patients, siblings and controls. Schizophr Res. 2020;223:135-147.
Richards AL, Pardiñas AF, Frizzati A, Tansey KE, Lynham AJ, Holmans P, Legge SE, Savage JE, Agartz I, Andreassen OA, Blokland GAM, Corvin A, Cosgrove D, Degenhardt F, Djurovic S, Espeseth T, Ferraro L, Gayer-Anderson C, Giegling I, van Haren NE, Hartmann AM, Hubert JJ, Jönsson EG, Konte B, Lennertz L, Olde Loohuis LM, Melle I, Morgan C, Morris DW, Murray RM, Nyman H, Ophoff RA; GROUP Investigators; van Os J; EUGEI WP2 Group; Schizophrenia Working Group of the Psychiatric Genomics Consortium; Petryshen TL, Quattrone D, Rietschel M, Rujescu D, Rutten BPF, Streit F, Strohmaier J, Sullivan PF, Sundet K, Wagner M, Escott-Price V, Owen MJ, Donohoe G, O'Donovan MC, Walters JTR. The Relationship Between Polygenic Risk Scores and Cognition in Schizophrenia. Schizophr Bull. 2020;46(2):336-344. doi: 10.1093/schbul/sbz061.
Werner MCF, Wirgenes KV, Haram M, Bettella F, Lunding SH, Rødevand L, Hjell G, Agartz I, Djurovic S, Melle I, Andreassen OA, Steen NE. Indicated association between polygenic risk score and treatment-resistance in a naturalistic sample of patients with schizophrenia spectrum disorders. Schizophr Res. 2020;218:55-62. doi: 10.1016/j.schres.2020.03.006.
Thank you for your thoughtful comments. We added the following text in the small “Future directions” section:
Given that all our associations were sex-specific, we also recommend separate analyses for males and females in schizophrenia, or controlling the results for patient’s sex, especially in relation with studies addressing COMT. Our results add to the current knowledge on sexually dimorphic influence of COMT gene upon psychiatric phenotypes (Harrison and Thunbridge, 2008), that was also reported in healthy individuals (Jiang et al, 2015). In addition, it would be interesting to explore how different COMT polymorphisms and haplotypes relate to psychopathology in different stages throughout the course of schizophrenia in males and females.
Harrison PJ, Tunbridge EM. Catechol-O-methyltransferase (COMT): a gene contributing to sex differences in brain function, and to sexual dimorphism in the predisposition to psychiatric disorders. Neuropsychopharmacology. 2008;33(13):3037-45.
Jiang W, Shang S, Su Y. Genetic influences on insight problem solving: the role of catechol-O-methyltransferase (COMT) gene polymorphisms. Front Psychol. 2015;6:1569. doi: 10.3389/fpsyg.2015.01569.
Finally, if you deem it relevant, a quick discussion of the evolutionary advantage of these COMT variants – for example in the context of the evolutionary paradox of schizophrenia – would be fascinating in order to again frame the research within a broader, more philosophical molecular evolutionary framework.
Answer: We have accepted this comment. We have added text in the Discussion “It is intriguing to put the COMT gene into evolutionary perspective. Given the role of prefrontal cortex in complex cognitive functioning, and the role of COMT enzyme in modulating prefrontal dopamine levels, COMT functional polymorphisms may have impacted adaptation to new environments during human evolution. Of note, while G allele of COMT rs4680 is the ancestral allele, while A allele is a result of mutation, which is considered advantageous for human evolution because A allele was unique to humans and related to higher intelligence (Piffer, 2013) and better insight problem solving (Jiang et al, 2015). Schizophrenia is very rare in societies with hunter-gatherer lifestyles (Rantala et al, 2022). In turn, hunter-gatherers had lower frequency of A allele than farming or industrialized populations (Piffer, 2013). While it is unknown if COMT gene has any role in the evolutionary paradox in schizophrenia, we speculate that the relation between higher-activity COMT polymorphisms and haplotype with lower intensity of some symptoms in schizophrenia may suggest distinct impact of COMT genes on schizophrenia onset and schizophrenia severity.
Piffer, D. Correlation of the COMT Val158Met polymorphism with latitude and hunter-gather lifestyle suggests culture–gene coevolution and selective pressure on cognition genes due to climate. Anthropological Science 2013, 121(3), 161–171, 2013
Jiang W, Shang S, Su Y. Genetic influences on insight problem solving: the role of catechol-O-methyltransferase (COMT) gene polymorphisms. Front Psychol. 2015;6:1569. doi: 10.3389/fpsyg.2015.01569.
Rantala MJ, Luoto S, Borráz-León JI, Krams I. Schizophrenia: The new etiological synthesis. Neurosci Biobehav Rev. 2022;142:104894. doi: 10.1016/j.neubiorev.2022.“
Comments on the Quality of English Language
Very minor edits to the English language can be made to improve the manuscript, throughout.
For example, line 72, “One of the most frequently investigated genes, associated with cognitive phenotypes, is a COMT gene” should read “One of the most frequently investigated genes, associated with cognitive phenotypes, is the COMT gene.”
Answer: Thank you, we have corrected this in the text.
As another example, line 258,
“Both COMT polymorphisms showed association with the PANSS general scores” should read “Both COMT polymorphisms showed an association with the PANSS general scores”.
Answer: Thank you, we have corrected this in the text.
25+ examples can be found throughout the text.
Answer: Replacements have been done.
